# Effect of Pit Mud on Bacterial Community and Aroma Components in Yellow Water and Their Changes during the Fermentation of Chinese Strong-Flavor Liquor

**DOI:** 10.3390/foods9030372

**Published:** 2020-03-23

**Authors:** Zhanzheng Gao, Zhengyun Wu, Wenxue Zhang

**Affiliations:** 1College of Biomass Science and Engineering, Sichuan University, Chengdu 610065, China; gzz126@126.com (Z.G.); wuzhengyun@scu.edu.cn (Z.W.); 2Sichuan Shuijingfang Co., Ltd., Chengdu 610037, China

**Keywords:** Chinese strong-flavor liquor, Yellow water, bacterial community, pit mud

## Abstract

As the main by-product of Chinese strong-flavor liquor, yellow water plays an important role in the formation of flavor components. Yellow water from different fermentation periods (30th day, 45th day, 60th day) was selected to analyze the aroma components by Headspace solid phase micro-extraction Gas Chromatography–Mass Spectrometry, and the microorganism community was evaluated by high-throughput sequencing technology and bioinformatics analysis of DNA. As the fermentation time was prolonged, the main flavor components significantly increased, and the amount of the common microbial population between yellow water and pit mud increased gradually. Among the common microorganisms, *Lactobacillus* accounted for the largest proportion, at about 56.96%. The microbes in the yellow water mainly belonged to Firmicutes. The abundance of Bacilli (the main bacteria) gradually decreased with time, at 87.60% at the 30th day down to 68.87% at the 60th day, but Clostridia gradually increased from 10.29% to 27.48%. At the genus level, some microbes increased significantly from the 30th day to 60th day, such as *Caproiciproducens,* which increased from 2.65% to 6.30%, and *Sedimentibacter*, increasing from 0.47% to 2.49%. RDA analysis indicated that the main aroma components were positively correlated with Clostridia and negatively correlated with Bacilli.

## 1. Introduction

Chinese strong-flavor liquor (CSFL) is the most important white liquor in Chinese liquor, accounting for more than half of the liquor market [1]. The key to determining the style of CSFL is to use mud pits for a long period of solid-state fermentation, generally 60 to 70 days per cycle, or even longer [2]. For example, the double-bottom fermentation process, which is important for improving the quality of CSFL [3,4], places the *Zaopei* (fermented grains, generally being composed of sorghum, rice, glutinous rice, corn, and wheat in proportion) at the bottom of the pit for a long time; i.e., for about two fermentation periods or even longer. During this process, the *Zaopei* is always in contact with the pit mud and yellow water, and large amounts of aromas are produced. Then, high-quality flavored wine can be produced by this *Zaopei*. In this process, yellow water and pit mud play an important role in producing aroma components. Yellow water is one of the main by-products in the production of CSFL. It is mainly formed in the alcohol fermentation period (according to the current production experience, the alcoholic fermentation stage is mainly carried out in the first 30 days of CSFL) [5]. In the process of alcohol fermentation, with the consumption of starch, the water contained in it and the water produced by fermentation flow out, and gradually flow down to the bottom of the pit to form yellow water. During this process, some nutrients and microorganisms in the middle and upper layers of fermented grains flow into the lower layer with the water and gradually sink to the bottom. According to reports, yellow water is rich in organic acids, alcohols, and esters. Many of these substances are aroma components of CSFL and are also important nutrients for microbial growth [6]. In addition, the yellow water is in contact with the mud and the fermented grains at the bottom of pit for a long time. Therefore, in addition to the microorganisms from *Zaopei*, the yellow water also takes microorganisms from the mud. These microorganisms are very important for the production of white wine aroma. In actual production, we usually use yellow water to improve the quality of CSFL according to production experience, which has obvious effects on improving the quality of wine.

High throughput sequencing technology has been widely used in the study of microorganisms in pit mud and *Zaopei* of CSFL [7,8,9]. It is often used to analyze the influence of microbial diversity and community structure on liquor quality in brewing process. Shannon index and Chao1 index are usually used to reflect the richness and diversity of microorganisms. The higher Shannon index and Chao1 index are, the higher microbial richness and diversity are [10,11]. The community structure can reflect the abundance of beneficial microorganisms in the pit mud and *Zaopei*. Through some statistical analysis methods, such as redundancy analysis (RDA) and Pearson correlation heatmap, correlation between environmental factors and microorganisms can be analyzed [9]. Therefore, we can use high-throughput sequencing to study the effect of pit mud on microorganisms and their changes in yellow water.

The main difference between Chinese liquor and other distilled liquors lies in the aroma of lower fatty acid ethyl esters and the high total acids content. CSFL mainly reflects the compound aroma dominated by ethyl caproate (EC). Other aroma components such as ethyl acetate (EA), ethyl lactate(EL), ethyl butyrate(EB), are also important aroma contributors, and organic acids, such as caproic acid(CA), acetic acid(AA), butyric acid(BA) and so on, are the main acids of CSFL, giving it soft mouthfeel and long aftertaste [12,13,14]. There are a lot of these acids and esters in yellow water [6]. Because of these organic acids, yellow water usually has higher acidity and lower pH value. These two values are often used to judge the quality of yellow water [15].In the lower part of pit, the fermented grains were immersed in yellow water for a long time. The content of these aroma components in yellow water could reflect their content in fermented grains to some extent [16]. The flavor components are distilled into liquor by distillation of fermented grains, which affect liquor quality. The content and proportion of these aroma components determine the flavor type and quality of the wine [17,18]. Sometimes experienced engineers can judge whether there is some problem in production according to the content and proportion of these aroma components.

The role of mud and its microorganisms in CSFL has been reported in a great deal of literature [9,19], but the function of yellow water and its microbes is not clear. In order to know how the microbes change in this process, how the microbes in the yellow water interact with microbes in the mud, and how they contribute to the formation of aromas, the changes of the bacterial community and main flavor components in the yellow water at different stages of late fermentation of CSFL and their relationship are studied in this paper. By this study, the theoretical basis will be laid for improving the quality of liquor by using yellow water, and new opportunities may be found for the comprehensive utilization of yellow water, which can improve the rate of high quality liquor and save cost for liquor factory.

## 2. Materials and Methods

### 2.1. Sampling Method for Yellow Water and Pit Mud

Yellow water: We selected three adjacent pits in a famous winery in Sichuan (the age of the pits is 25 years) and pre-buried a stainless steel pipe (5 cm in diameter, with a large number of very small holes at the bottom) before putting fermented grains into the pits (length × width × dept: 3 m × 2 m × 1.8 m, thickness of pit mud: about 10 cm). The upper mouth of the pipe was sealed before sampling. On the 30th day, 45th day, and 60th day of fermentation time, the sealing port was opened, and about 6 L of yellow water was pumped from the bottom of each pit (Appendix A).

Pit mud: After the fermentation was completed and the fermented grains were taken out, pit mud was taken from two different sites (one site was on the pit wall and the other site on the pit bottom, near the yellow water sampling point in each pit). About 100 g of mud was taken from each point, and then mixed. Then, the mud was placed in liquid nitrogen and stored for use.

### 2.2. Determination of pH and Acidity of Yellow Water

We took 50 mL of yellow water in a 100 mL beaker, placed the handheld pH meter in the solution and let it stand to determine the pH. No change occurred within half a minute. We read the value and measured each sample three times.

Take 1 mL of yellow water mixed evenly and add it into a 250 mL conical flask, add 50 mL of water, titrate with 0.1 mol/L NaOH (add 2 drops of phenolphthalein indicator first) to a reddish color, and take half a minute as the end point. We measured each sample three times.

Acidity: for every 1 mL of yellow water, 1 mL of 0.1 mol/L NaOH solution for titration is 1 degree.

Acidity = CV/0.1, C, NaOH solution concentration, mol/L; V, NaOH solution consumed in titration, mL; 0.1, Standard NaOH solution, mol/L.

### 2.3. Flavor Components of Yellow Water and Pit Mud

Headspace solid phase micro-extraction (HS-SPME) was performed for the extracted trace volatiles from yellow water and pit mud.

We took 5 mL of yellow water in a headspace bottle, added 2 g of NaCl and 100 μL of the internal standard butyl acetate (6.8 mg/mL), equilibrated the mixture at 70 °C for 10 min, then extracted it by headspace adsorption for 35 min. Then the headspace needle was inserted into sample inlet of the gas chromatography–mass spectrometer (GS-MS), and the sample was injected at 230 °C, and the fragrance components were analyzed [20].

We took 2 g of mud in a headspace bottle, added 5 mL of deionized water and 2 g of NaCl, added 100 uL of the internal standard substance butyl acetate (8.4 mg/mL), and extracted the mixture for 30 min with the ultrasonic wave (Scientz IID ultrasonic cell grinder, power, 250 W, temperature, 25 °C). The mixture was equilibrated at 70 °C for 10 min, then extracted it by headspace adsorption for 35 min. Then the headspace needle was inserted into sample inlet of the GS-MS, and the sample was injected at 230 °C, and the fragrance components were analyzed [21].

The volatile component was extracted with a 50/30 μm Divinylbenzene/Carboxen/Polydimethylsiloxane (DVB/CAR/PDMS) extraction head (Supelco, Inc., Bellefonte, PA), which was aged at 270 °C for 1 h to remove impurities. The instrument was MPS2 (Multipurpose sampler, automatic SPME) from Gerstel. The concentration was calculated by the percentage of the peak area.

GC-MS: Agilent, 7890A-5975C.

Conditions of GC [22]: The capillary column was DB-Wax (30 m × 0.25 mm × 0.25 μm). The inlet and detector temperatures were both 230 °C, the carrier gas was He, and the flow rate was 2 mL/min. Temperature-programmed route: the starting temperature was 40 °C, which was held for 2 min, then increased to 230 °C at a rate of 4 °C/min, which was maintained for 15 min.

Conditions of MS [22]: Interface temperature of chromatography and mass spectrometry was 230 °C; Ionization mode was electron bombardment source (EI); Detection mode was full scan, Ionization energy was 70 ev; Mass range was 30–500 amu; Acquisition frequency was 0.1 s/scan; Ion source temperature was 200 °C and detector voltage was 1450 v. The database for mass spectrometry was Wiley and the National Institute of Standards and Technology (NIST) library.

Each sample was repeated three times.

### 2.4. DNA Extraction of Yellow Water and Pit Mud

The fresh yellow water was centrifuged at 13,000× *g* for 10 min (ZONKIA, HC-3806), the supernatant was decanted, and the cell pellet was collected. Then, the precipitate was taken out, and the total genomic DNA of the yellow water microbes was extracted by the FastDNA SPIN Kit for soil. The DNA extraction of pit mud microorganisms was performed using the DNeasy Power-Soil Kit. The extraction method was carried out in accordance with the experimental instructions provided by the reagent company. The extracted microbial DNA was subjected to mass detection by agarose electrophoresis, and the library was constructed and sequenced after passing the test.

### 2.5. High-Throughput Sequencing Analysis

The construction and sequencing were completed with a Shanghai Majorbio Biotechnology Co., Ltd. High-throughput sequencing targeted the V3–V4 region of the bacterial 16S rRNA gene, and the amplification primers were 338F (5′-ACTCCTACGGGAGGCAGCAG-3′) and 806R (5′-GGACTACHVGGGTWTCTAAT-3′). The sequencing platform was Illumina MiSeq. The original sequence of sequencing was spliced and quality-controlled by FLASH and Trimmomatic software. The quality of the sequence was as follows: (1) we removed the bases with a tail value of 20 or less, and removed the sequences below 50 bp after quality control; (2) the maximum number of primer mismatches was 2, the number of mismatches allowed by the barcode was 0, and the fuzzy bases were removed. (3) The minimum overlap sequence length was 10 bp, and the overlap area of the spliced sequence allowed a maximum mismatch ratio of 0.2. Chimer sequences were removed using Uchime software. Usearch (version 7.0, http://drive5.com/uparse/) was used to cluster the sequences according to 97% similarity. Then, we performed OTU (operational taxonomic units), extracted representative sequences, and obtained OTU tables. The bacterial OTU representative sequence was aligned with the Silva (Release 132, http://www.arb-silva.de) database for taxonomic annotation. The bacterial flora heatmap (Pearson Correlation Heatmap) and RDA were analyzed by the R language (vegan package).

## 3. Results

### 3.1. pH and Acidity of Yellow Water

As the fermentation time was prolonged, the pH of the yellow water gradually decreased from 3.60 at the 30th day to 3.44 at the 60th day, but the decrease was small. The acidity was 6.07 (mL Standard NaOH solution) at 30 days, which increased to 6.70 (mL Standard NaOH solution) at 60 days (Appendix A).

### 3.2. The Aroma Components of the Pit Mud and Yellow Water

The aroma components of the pit mud and the changes of the main aroma components of the yellow water at 30 days, 45 days, and 60 days were measured by the HS-SPME-GC-MS method. The absolute content of all these components at 60 days was higher than that at 30 days (Appendix A). Some substances had increased; for example, AA increased from 20.01 mg/L to 261.78 mg/L, BA increased from 27.38 mg/L to 126.89 mg/L, and EL increased from 25.43 mg/L to 181.66 mg/L. At 30 days, CA amounted to 564.33mg/L, while this value was 457.5 mg/L at 45 days and 960.01 mg/L at 60 days. EC showed the same trend; it was 446.08 mg/L at 30 days, and reached 585.38 mg/L at 60 days. Some ingredients first decreased slightly and then increased. For example, EB was 17.69 mg/L at 30 days, then decreased to 16.70 mg/L at 45 days, and increased to 21.45 mg/L at 60 days. According to the relative content of aroma components (Figure 1), AA, BA, and EL increased significantly, while AA increased most obviously, from 1.28% to 8.5%. The relative contents of EA, EB and EC gradually decreased. CA first dropped and then rose. EC and CA were always the two highest proportion components. At 30 days, the CA percentage reached 53.93%, and EC reached 41.89%. CA accounted for 44.15% at 60 days, and EC was 26.92%. The two components were the main components in yellow water, which was similar to the pit mud, where CA reached 5059.93 mg/kg and the EC content was 947.09 mg/kg. It could be seen that, during the long-term solid-state fermentation process, prolonging the fermentation time was beneficial to the improvement of the flavor component in the yellow water. These aroma components could then enter the fermented grains through the yellow water and were then distilled into the wine.

### 3.3. Diversity of Bacterial Community in Yellow Water and Pit Mud

The sequencing of the yellow water (nine samples) and pit mud (three samples) resulted in 477,218 qualified sequences with an average length of 418.69 bp for all samples, and samples contained 30,381–52,008 sequences (Table 1 and Appendix A). Overall, 501 different OTUs (97% similarity) from all 12 samples were obtained.

The Shannon index of yellow water gradually increased with the increase of fermentation time, from 1.31 ± 0.05 at 30 days to 2.58 ± 0.35 at 60 days, but this was less than the 2.86 ± 0.24 for the mud. The yellow water Chao1 index also increased with time, from 362.81 ± 23.10 to 418.72 ± 35.66, while the mud Chao1 was 294.78 ± 31.33 and lower in all yellow water samples (Table 1).

According to the Shannon index, the bacterial diversity in yellow water increased with time, but the diversity of mud was the highest. The yellow water Chao1 index increased with time, and the index of pit mud was the lowest. This indicated that the number of OTUs increased gradually and the species richness increased, which was consistent with the description of the Shannon index. However, the Chao1 index in the mud was smaller than that in the yellow water, and the Shannon index indicated that the bacteria in the mud was richer than the yellow water, which indicated that the rare OTU in the yellow water was increased, and the rare OTU in the mud was decreased.

Yellow water had an average of 153 genera at 30 days, 161 genera at 45 days, and 174 genera at 60 days. With the prolongation of fermentation time, the microbial population of yellow water increased and was greater than the pit mud (137 genera). There were 143 common genera between HS30, HS45 and HS60, which indicated that the microbial population of yellow water increased from 30 days to 60 days, but there was little increase in the number of increased microorganisms, and the common genera microorganisms made up the majority. When the yellow water was in contact with the mud for a long time, the microorganisms were definitely affected by the mud. The common genus number between the yellow water and pit mud was 98 (Figure 2A). With the prolonged contact time between the yellow water and the mud, there were more and more common bacteria in them. At 30 days, there were 103 genera between yellow water and mud, and the common number of microorganisms increased to 118 at 60 days.

Among the bacteria shared by pit mud and yellow water, Lactobacillus accounted for the most, at about 56.96%. Other common bacteria were *Sedimentibacter*, 9.76%, *Syntrophaceticus*, 7.14%, *Caproiciproducens*, 3.60%, *Hydrogenispora*, 3.12%, *g_norank_f_Peptococcaceae*, 2.69%, *g_norank_f_ Clostridiaceae_1*, 1.75% (Figure 2B).

### 3.4. Changes of Bacterial Community

The community distribution of microbes at the phylum level of yellow water and pit mud is shown in Appendix A. Firmicutes in the yellow water and pit mud accounted for the majority, at 97.79% at 30 days, 97.65% at 45 days, 96.48% at 60 days and 98.91% in pit mud. The proportion of Firmicutes microorganisms in the yellow water decreased slightly with time, while the proportion of Bacteroidetes increased gradually, from 0.67% at 30 days to 1.40% at 60 days.

At the class level, the main bacteria in the yellow water were Bacilli, making up 87.60% at 30 days, 79.44% at 45 days, and 68.87% at 60 days. The abundance gradually decreased with prolonged time. Then, Clostridia gradually increased in the yellow water with time, increasing from 10.29% at 30 days to 27.48% at 60 days. However, in pit mud, Clostridia accounted for 97.44% and Bacilli only 1.28%. (Figure 3A).

At the genus level, the main bacteria in yellow water were *Lactobacillus*, amounting to 87.02% at 30 days, 78.59% at 45 days, and 67.81% at 60 days. Other dominant microorganisms included *Sedimentibacter, Syntrophaceticus, Caproiciproducens, Hydrogenispora, Clostridium_sensu_stricto_15, norank_f_Clostridiaceae_1, Alkalibaculum,* etc. These microorganisms increased with time, among which *Sedimentibacter, Caproiciproducens, norank_f_Clostridiaceae, Alkalibaculum* increased significantly, especially *Caproiciproducens*, which increased from 2.65% at 30 days to 6.30% at 60 days, *Sedimentibacter* increased from 0.47% at 30 days to 2.49% at 60 days, *norank_f_Clostridiaceae* from 1.56% at 30 days to 2.95% at 60 days, and *Alkalibaculum* increased from 0.95% at 30 days to 2.56% at 60 days. The main dominant bacteria in the mud were *Sedimentibacter* (30.81%), *Syntrophaceticus* (23.77%), *Hydrogenispora* (10.51%), *norank_f-_Peptoccoccaceae* (9.16%), *Caloribacterium* (5.01%), *unclassified_f_Lachnospiraceae* (1.86%), *Tissierella* (2.23%), *norank_f_Family_XIV* (2.09%), *Clostridium_sensu_stricto_12* (1.53%), *norank_f_Lachnospiraceae* (1.11%). At 60 days, the dominant bacteria with a higher abundance in yellow water than in pit mud were *Lactobacillus, Caproiciproducens, norank_f_Clostridiaceae_1, Alkalibaculum*, while those with higher abundance in pit mud were *Sedimentibacter, Syntrophaceticus, Hydrogenispora, norank_f-_Peptoccoccaceae, Caloribacterium, unclassified_f_Lachnospiraceae,* etc. In particular, *Caproiciproducens* amounted to only 0.55% in the mud, but 6.30% in the yellow water. The absolutely dominant microorganisms in yellow water, *Lactobacillus*, amounted to only 1.25% in pit mud. It can be seen from the above that the structure of the main bacteria in the yellow water and pit mud was similar, but the abundance was very different (Figure 3B).

### 3.5. Relationship between Bacterial Community and Aroma Components of Yellow Water

RDA was conducted using abundant microbes, together with flavor components and pH (Figure 4). Species-environment correlations for both axes were higher than 99%, suggesting the remarkable correlation between bacterial Community structure and environmental factors. At the class level, CA, EC, EA, AA, BA, EA, EL were positively correlated with Clostridia. The formation of these aroma components was closely related to Clostridia, but negatively correlated with Bacilli, which was not conducive to the formation of these substances in yellow water. The pH was positively correlated with Bacilli and negatively correlated with Clostridia.

The top 20 genera were selected, and Pearson correlation coefficient of these species and aroma components was analyzed to form the heatmap (Figure 5). From the Pearson heatmap, the contents of CA and EC showed the strongest positive correlation with *Syntrophomonas*. Other positive correlations were *Caloribacterium*, *Sedimentibacter, Syntrophaceticus, Clostridium_sensu_stricto_12, Fermentimonas, Caldicoprobacter, Tissierella, Tepidimicrobium*, followed by microorganisms such as *Caproiciproducens*. EA, AA, BA, and EL had the strongest positive correlation with *Syntrophomonas, Caloribacterium*, and *Sedimentibacter*, followed by *Caproiciproducens Fermentimonas, Caldicoprobacter, Tissierella, Tepidimicrobium*. The strong correlation with EB was *norank_f_Anaerolineaceae, Syntrophomonas*. *Lactobacillus* was negatively correlated with the formation of these substances, and the most negative correlation was AA, followed by BA, EL, CA, EC, then EB, EA.

## 4. Discussion

The yellow water came from the fermented grains and was also in contact with the mud for a long time, so the substances and microorganisms in it were affected by both the grains and pit mud. EC was the main characteristic aroma component of CSFL [23]. CA was the necessary acid for the synthesis of EC. From 30 days to 45 days, the change of EC and CA was not obvious. However, the absolute content of CA and EC increased significantly in the later period, and both always accounted for a high proportion in the aroma components of yellow water, which was beneficial to increasing the contents of the two in the fermented grains. This indicated that the long-term contact between fermented grains and yellow water was beneficial for increasing the main flavor components and improving the quality of the liquor. This also reasonably explained the rationality of using double-bottom technology to prolong the fermentation period to produce flavored wine.

The common synthesis method of CA was that microorganisms perform synthesis by the chain extension reaction of ethanol and lactic acid. AA and BA needed to be consumed when ethanol was used as a substrate, and AA and BA also needed to be used when lactic acid was used as substrate [24,25,26]. After about 30 days, alcoholic fermentation had been basically completed and the content of ethanol should be relatively high. At this time lactic acid in fermented grains and yellow water was also higher [27]. And so, microorganisms were able to use ethanol, lactic acid and AA to synthesize CA. Thus, the content of CA in yellow water was higher, and the content of AA and BA was low. Although the content of CA increased at 45 days, the increase was not large, which might be related to the increase of fatty acid oxidizing bacteria, while the content of AA and BA increased rapidly. In the later stage, the production of AA and BA increased more obviously; in particular, AA increased by ten times, which also promoted the synthesis of CA, and its content also increased considerably. These acids were organic acids with strong buffering capacity, and their increase increased the acidity from 5.90(mL Standard NaOH solution) to 6.70(mL Standard NaOH solution), but the pH was only reduced from 3.60 to 3.44.

The Shannon and Chao1 indices of microorganisms in yellow water increased gradually with time, which indicated that the species and quantity of microorganisms in yellow water increased. However, the Shannon index in yellow water was smaller than that in pit mud, while the Chao1 index was larger than that in pit mud, indicating that the number of microorganisms in yellow water was less than that in pit mud, but the species were increased compared to in pit mud. This was consistent with the results in the Venn diagram (Figure 2A). In the Venn diagram, the genera and species in pit mud were smaller than those in yellow water, and the population in yellow water was increasing gradually. This might be because yellow water had been in contact with pit mud and fermented grains for a long time. In addition to the microorganisms from pit mud, there were also a large number of microorganisms from grains. At the same time, yellow water was also affected by pit mud, and the common microbial species of yellow water and pit mud were gradually increasing. At the genus level, common microorganisms increased from 103 to 118, which indicated that the longer the contact time, the more bacteria in the pit mud entered yellow water. Apart from *Lactobacillus*, which accounted for the largest proportion, other microorganisms were *Sedimentibacter, Syntrophaceticus, Caproici producers* and *Hydrogenispora*. These microorganisms were closely related to the flavor components in yellow water and were positively related. With the prolongation of the contact time between the yellow water and pit mud, the abundance of bacteria shared by yellow water and pit mud increased gradually, indicating that the reproduction and metabolism of these microorganisms promoted the formation of aroma components.

The abundance of Firmicutes in the bacterial community of yellow water and pit mud occupied an absolute dominance, which was consistent with the reported viewpoints in other literature [6,28], but the abundance ratio was different. In this study, the abundance of Firmicutes in yellow water and pit mud was more than 90%, which was much higher than other bacteria. Bacilli was always dominant in yellow water, but with the prolongation of contact time with pit mud, the proportion of Bacilli gradually decreased. In pit mud, the dominant Clostridia gradually increased in yellow water. Clostridia was the main microorganism producing CA in pit mud [9,29]. In the RDA analysis, the correlation between Clostridia and aroma components was positive, which indicated that the bacteria in Clostridia played an important role in the formation of these aroma components, while Bacilli played an opposite role.

*Caproiciproducens, Sedimentibacter, Alkalibaculum,* and other microorganisms increased significantly with the prolongation of fermentation time. *Caproiciproducens*, isolated and identified as caproic acid-producing bacteria, belonged to *Clostridium IV* of *Ruminococcaceae* [30] and was able to synthesize CA from lactic acid. Lactic acid was abundant in yellow water. The bacteria could use lactic acid to synthesize CA. As time went on, the abundance gradually increased, which was beneficial to the formation of CA. However, in pit mud, the abundance was relatively small. *Sedimentibacter* could metabolize amino acids to produce organic acids, such as AA and BA [31]. *Sedimentibacter* was the most abundant microorganism in pit mud. It was also a representative microorganism in old pit mud [32]. In the late stage of fermentation, yeast died in large quantities and produced a large number of amino acids flowing into yellow water. *Sedimentibacter* could use these amino acids to reproduce and metabolize and produce a large number of organic acids. The Pearson Heatmap also showed that *Sedimentibacter* was positively correlated with these aroma components. *Alkalibaculum* could use H_2_ and CO_2_ to metabolize AA, ethanol and so on [33]. Its abundance in yellow water was much higher than in pit mud. In the later stage of fermentation, H_2_ and CO_2_ were present in the pit, which might promote reproduction in yellow water; the increase of AA content might be related to this. *Alkalibaculum* was also positively related to AA in the Pearson thermogram. *Syntrophomonas* could metabolize H_2_ and CO_2_ using long-chain fatty acids as substrates [34], which were strongly positively correlated with various aroma components. Although not dominant microorganisms in pit mud and yellow water, they could degrade long-chain fatty acids and provide suitable substrates for microorganisms such as methanogens, which might promote carbon cycling in pits. *Syntrophaceticus* was a very different in pit mud and yellow water. It belonged to the Firmicutes–Clostridia class, and exhibited an acetic acid oxidation ability in co-culture with methanogens [35]. AA was necessary for the synthesis of EA. The degradation of AA might help to reduce the synthesis of EA, which provided an opportunity to solve the problem of excessive EA content in liquor production.

## 5. Conclusions

The main aroma components in yellow water, such as caproic acid (CA), acetic acid (AA), butyric acid (BA), ethyl caproate (EC), and ethyl lactate (EL) increased with the prolongation of fermentation time. At the same time, the microorganisms in the yellow water were obviously affected by the pit mud. With the prolonged contact time between the yellow water and the mud, bacteria were increasingly common. Some microorganisms that were beneficial to the formation of aroma components increased, such as Clostridia, increasing from 10.29% at 30 days to 27.48% at 60 days, while those that were not conducive to the formation of aroma components decreased, such as Bacilli, which amounted to 87.60% at 30 days, 79.44% at 45 days, and 68.87% at 60 days.

## Figures and Tables

**Figure 1 foods-09-00372-f001:**
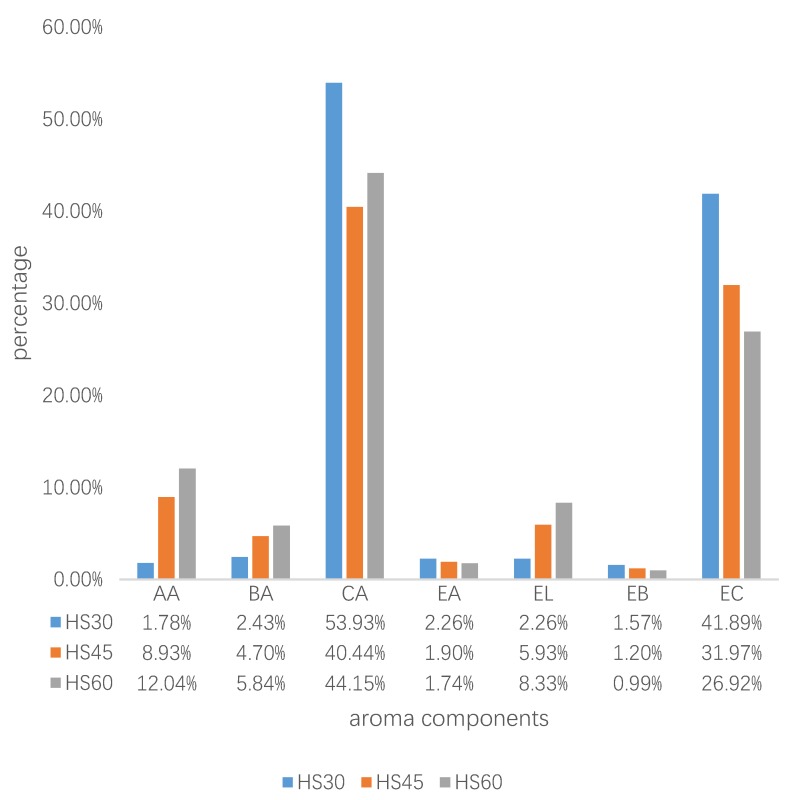
Percentage of aroma components at different fermentation times. AA: acetic acid, BA: butyric acid, CA: caproic acid, EA: ethyl acetate, EL: ethyl lactate, EB: ethyl butyrate, EC: ethyl caproate. HS30, yellow water at 30th day, HS45, yellow water at 45th day, HS60, yellow water at 60th day.

**Figure 2 foods-09-00372-f002:**
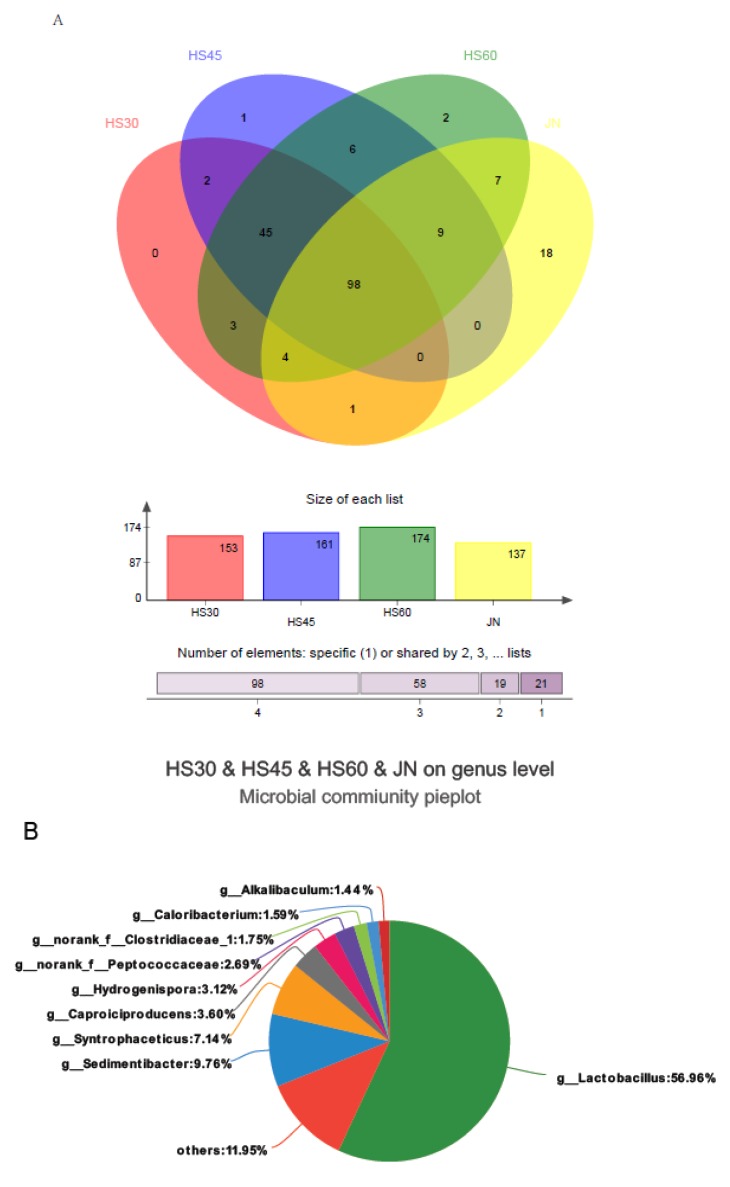
Common microbes between yellow water and pit mud. (**A**) Venn diagram of yellow water and pit mud, (**B**) Percentage of common microbes between yellow water and pit mud (at the genus level).

**Figure 3 foods-09-00372-f003:**
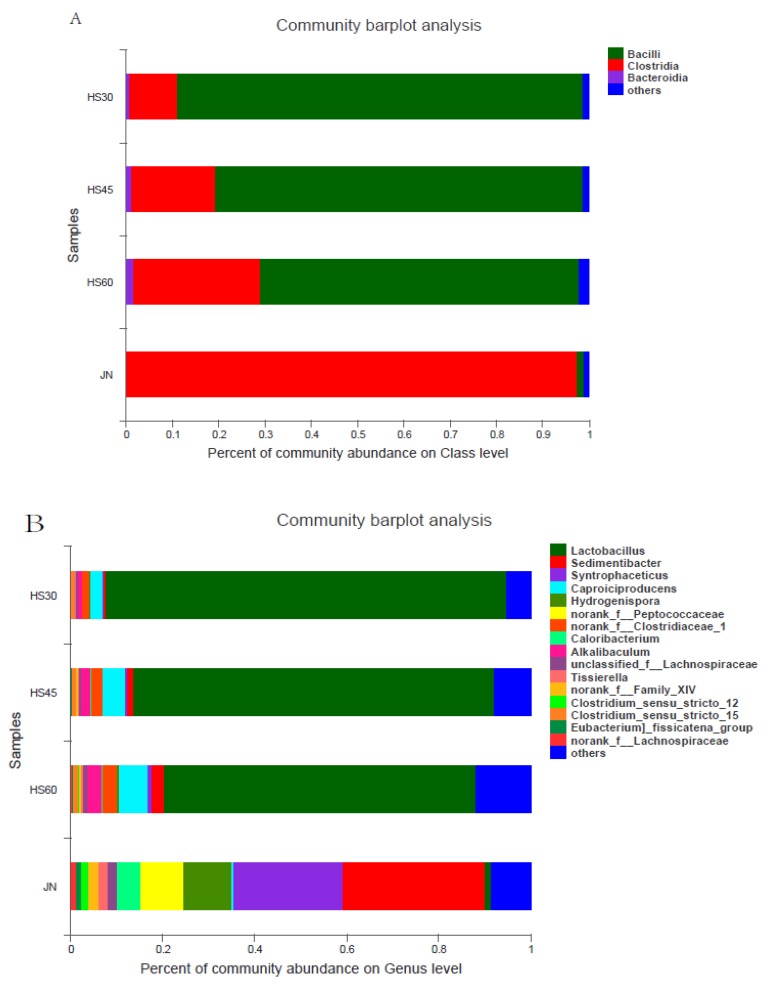
Percent of community abundance. (**A**) Percentage of community abundance at the class level; (**B**) Percentage of community abundance at the genus level. HS30: yellow water on the 30th day, HS45: yellow water on the 45th day, HS60: yellow water on the 60th day, JN: pit mud.

**Figure 4 foods-09-00372-f004:**
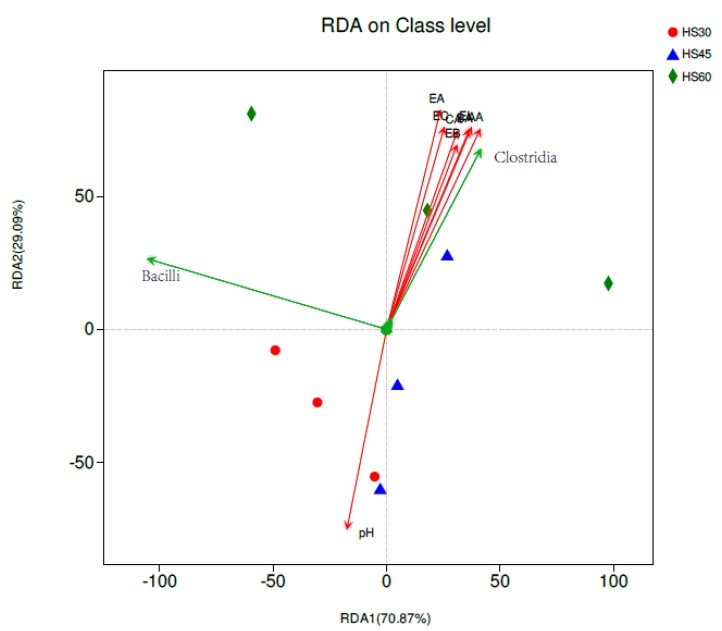
Redundancy analysis of the microbe community structure and aroma components. Arrows indicate the direction and magnitude of measurable variables associated with the microbe community structure.

**Figure 5 foods-09-00372-f005:**
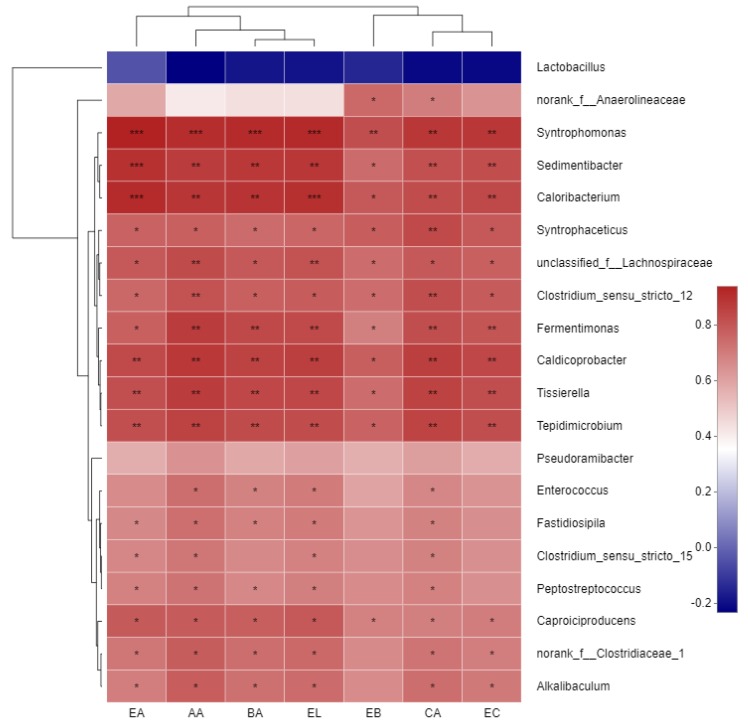
Pearson Correlation Heatmap of Microorganisms and aroma components. The *X*-axis and *Y*-axis are environmental factors and species, respectively, and R and *P* values are obtained by calculation. The R value is displayed in different colors in the graph. If the *P* value is less than 0.05, it is marked with a “*” sign. The legend on Table 0 < *P* < 0.05, ** 0.001 < *P* < 0.01, *** *P* < 0.001.

**Table 1 foods-09-00372-t001:** Sequence number and diversity indices calculated based on a cutoff of 97% similarity.

Sample	Sequence Number	Shannon	Chao1
HS30	35918.33 ± 4036.73	1.31 ± 0.05	362.81 ± 23.10
HS45	35285.67 ± 4621.46	1.87 ± 0.72	371.99 ± 37.31
HS60	42233.67 ± 8418.46	2.58 ± 0.35	418.72 ± 35.66
JN	45635 ± 5843.77	2.86 ± 0.24	294.78 ± 31.33

HS30: yellow water at 30th day, HS45: yellow water at 45th day, HS60: yellow water at 60th day, JN: pit mud.

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
