# Peer review of "Effect of Pit Mud on Bacterial Community and Aroma Components in Yellow Water and Their Changes during the Fermentation of Chinese Strong-Flavor Liquor"

_foods, 2020, doi:10.3390/foods9030372_

Round 1
Reviewer 1 Report
The manuscript „Changes of microbial community and aroma components in yellow water during the fermentation of Chinese strong-flavor liquor and the relationship with pit mud” has been much improved, however, Authors forgot, I think, to upload the supplementary files, and this is the main drawback. Without this data, I don’t have the complete view on this paper, especially that authors have made some changes in comparison to the previous version of this paper. Some minor issues:
- L61-62 – “Manufacturers are very concerned about these, and guide producing according to these.” This sentence is too general, please rewrite
- L 150-152 – the reference in the text to the appropriate table is needed (e.g. “the acidity … (Table S1)”
- L157-167 – same comment – there should be information in the text where these results can be found (e.g. “from 25.43mg/L to 181.66mg/L (Table S2)”
- L180 – please include the reference to supplementary materials too (e.g. “30,381–52,008 sequences (Table 1 and S4).”
- L 212-215 – is this supposed to be a figure caption?
Author Response
We appreciate the constructive suggestion of Reviewer 1. We are very sorry for the supplementary files. We addressed all the points raised by the reviewer, as summarized below.
- L61-62 – “Manufacturers are very concerned about these, and guide producing according to these.” This sentence is too general, please rewrite
We have rewritten this sentence.
‘Sometimes experienced engineers can judge whether there is some problem in production according to the content and proportion of these aroma components.’
See line 75-77.
- L 150-152 – the reference in the text to the appropriate table is needed (e.g. “the acidity … (Table S1)”
This has been revised. See line 170.
- L157-167 – same comment – there should be information in the text where these results can be found (e.g. “from 25.43mg/L to 181.66mg/L (Table S2)”
This has been revised. See line 174.
- L180 – please include the reference to supplementary materials too (e.g. “30,381–52,008 sequences (Table 1 and S4).”
This has been revised. See line 199.
- L 212-215 – is this supposed to be a figure caption?
This is not a figure caption and has been revised. See line231-234

Reviewer 2 Report
Comments to the manuscript:
Introduction
Line 54 ‘[…] lower fatty acid ethyl ester and the high total acid content.’ I think that these names should be expressed in plural, i.e. esters, acids.
Materials and Methods
Line 113 GC-MS: Agilent, 7890A- 5975C [16]. Why the authors indicate reference at the model of used GC apparatus? I think that is unnecessary because each laboratory works with its own apparatus.
Results
3.1. pH and Acidity of yellow water –
Lines 151-152 Unit of acidity should be placed at the values of acidity
3.3. Diversity of Microbial Community in Yellow Water and pit mud
Line 185 Give please the information what mean the Shannon and Chao1 indexes?
Line 187 The values of the indexes in the text should be the same as present in the Table 1, i.e. should include standard deviation
Line 226 (Fig 3.) replace with (Fig. 3A)
Line 246 (Fig 3.) replace with (Fig. 3B)
Discussion
Lines 292-293 ‘So the microorganisms could use ethanol, lactic acid and AA to synthesize CA. 292 At this time, microorganisms were able to use ethanol, lactic acid and AA to synthesize CA .’ Repeated sentences. Rearrange the text
Lines 299-300 Unit of acidity should be placed
Conclusion
Lines 356-358 The sentence ‘The main aroma components in yellow water increased with the prolongation of fermentation 356 time, such as caproic acid (CA), acetic acid (AA), butyric acid (BA), ethyl caproate (EC), and ethyl 357 lactate (EL).’
replace with
‘The main aroma components in yellow water, such as caproic acid (CA), acetic acid (AA), butyric acid (BA), ethyl caproate (EC), and ethyl lactate (EL) increased with the prolongation of fermentation time.’
Supplementary materials
Are all Tables and Figures needed? For instance the Tables S1, S2, S3 are not listed in the manuscript. Moreover, Table S3 contains probably the same data as the Table 1. I suggest to check if there are no repetitions in the contents of tables and figures placed in the manuscript and in supplementary materials.
Author Response
We would like to thank the Reviewer 2 for their thoughtful review of our manuscript. We believe that the additional changes we have made in response to the reviewer comments have made this a significantly stronger manuscript. Below is our point-by-point response to the comments.
Introduction
- Line 54 ‘[…] lower fatty acid ethyl ester and the high total acid content.’ I think that these names should be expressed in plural, i.e. esters, acids.
This has been revised. See line 65.
- Materials and Methods
Line 113 GC-MS: Agilent, 7890A- 5975C [16]. Why the authors indicate reference at the model of used GC apparatus? I think that is unnecessary because each laboratory works with its own apparatus.
When detected these parameters with this apparatus, we referred to the detection conditions in this reference. Now we have labeled this reference again. See lines 129, 133.
- Results
3.1. pH and Acidity of yellow water –
Lines 151-152 Unit of acidity should be placed at the values of acidity
Unit of acidity has been added. See lines168-169.
- 3. Diversity of Microbial Community in Yellow Water and pit mud
Line 185 Give please the information what mean the Shannon and Chao1 indexes?
The information of Shannon and Chao1 indexes has been added in the Introduction. Seen lines55-57.
- Line 187 The values of the indexes in the text should be the same as present in the Table 1, i.e. should include standard deviation
This has been revised. See lines 205-207.
- Line 226 (Fig 3.) replace with (Fig. 3A). Line 246 (Fig 3.) replace with (Fig. 3B)
This has been revised. See lines 245, 264.
- Discussion
Lines 292-293 ‘So the microorganisms could use ethanol, lactic acid and AA to synthesize CA. 292 At this time, microorganisms were able to use ethanol, lactic acid and AA to synthesize CA .’ Repeated sentences. Rearrange the text
This has been revised.
‘And so, microorganisms were able to use ethanol, lactic acid and AA to synthesize CA.’
See lines314-315.
- Lines 299-300 Unit of acidity should be placed
This has been revised. See lines321-323.
- Conclusion
Lines 356-358 The sentence ‘The main aroma components in yellow water increased with the prolongation of fermentation 356 time, such as caproic acid (CA), acetic acid (AA), butyric acid (BA), ethyl caproate (EC), and ethyl 357 lactate (EL).’
replace with
‘The main aroma components in yellow water, such as caproic acid (CA), acetic acid (AA), butyric acid (BA), ethyl caproate (EC), and ethyl lactate (EL) increased with the prolongation of fermentation time.’
This sentence has been replaced. See lines379-382.
- Supplementary materials
Are all Tables and Figures needed? For instance the Tables S1, S2, S3 are not listed in the manuscript. Moreover, Table S3 contains probably the same data as the Table 1. I suggest to check if there are no repetitions in the contents of tables and figures placed in the manuscript and in supplementary materials.
Thank you for your advice very much. We think these tables and Figures are needed and have put the Tables (S1-line170, S2-line174, S3-line199) where they're referenced. The same data has been deleted in Table S3.

Reviewer 3 Report
General comments: this article measure interesting parameters related to microorganism community and aroma and their interaction in a spirit drink. Title needs a word to include the idea of interaction between microorganism, aroma, pit mud and yellow water. The main objective is not clear, the authors should answer the question: What advances are providing this work to the spirits and wine industry? Statistical analysis is not described and only mentioned in the abstract (RDA). Considering the high number of aroma compounds in distilled liquor, why the authors have chosen only 6 aroma compounds? Abstract and/or introduction should mention that high-throughput sequencing analysis is a DNA technique. Some microbiologist are not confident with DNA techniques to identify microorganisms. Introduction is not justifying some measured parameters as pH and acid analysis.
Abstract:
Lines
15 Include DNA analysis.
18 Although RDA is a known statistical abbreviation, the authors were using other statistical tools. For example, pearson correlation map is not mentioned in the experimental section.
Keywords: I would substitute the word “Chinese strong-flavor liquor” by “white liquor” or “spirit drink”; similarly, other synonymous for “yellow water” should be instead included.
Introduction:
Lines
35 Large amounts of aromas is not associated with a better wine quality. This idea is also repeated in line 284
87-89 General objective should be rewritten.
Materials and methods:
The number of analyses and repetitions are not mentioned along this section in any parameter. Flavor components and their standards are not mentioned in this section.
Lines
112 Detection conditions are described from line 118
113 Reference 16 is only using the same chromatograph or is also applying GC conditions and/or MS conditions?
Results and discussion:
Lines
156 In my opinion, absolute contents of aromas are not relevant in this study.
Odor activity value should be a better approach to have an idea of the aromatic influence.
180 What do the authors mean with OTUs?
182 What means Shanon and Chao 1 indexes? They are briefly described.
Figures and tables:
Is possible to increase the letter size in Figure 5?
Author Response
We want to begin by thanking Reviewer 3 for writing that “This article measure interesting parameters related to microorganism community and aroma and their interaction in a spirit drink.” We believe that the additional changes we have made in response to the reviewers comments have made this a significantly stronger manuscript. Below is our point-by-point response to the comments.
- Title needs a word to include the idea of interaction between microorganism, aroma, pit mud and yellow water.
We have changed the title: ’Effect of pit mud on microbial community and aroma components in yellow water and their changes during the fermentation of Chinese strong-flavor liquor’. See lines 2-5.
The word ‘effect’ can reflect interaction between microorganism, aroma, pit mud and yellow water.
- The main objective is not clear, the authors should answer the question: What advances are providing this work to the spirits and wine industry?
We have rewritten the main objective. See lines 79-86.
In order to know how the microbes change in this process, how the microbes in the yellow water interact with microbes in the mud, and how they contribute to the formation of aromas, the changes of the microbial community and main flavor components in the yellow water at different stages of late fermentation of Chinese strong-flavor liquor and their relationship are studied in this paper. By this study, the theoretical basis will be laid for improving the quality of liquor by using yellow water, and new opportunities may be found for the comprehensive utilization of yellow water, which can improve the rate of high quality liquor and save cost for liquor factory.
- Statistical analysis is not described and only mentioned in the abstract (RDA).
We have added some content about statistical analysis, in Introduction, Materials and Methods, Result and Discussion.
See lines 58-60, 163, 272-274, 283-284.
- Considering the high number of aroma compounds in distilled liquor, why the authors have chosen only 6 aroma compounds?
The main difference between Chinese liquor and other distilled liquors lies in the aroma of lower fatty acid ethyl esters and the high total acid content. Chinese strong-flavor liquor mainly reflects the compound aroma dominated by ethyl caproate. Other aroma components such as ethyl acetate, ethyl lactate, ethyl butyrate, are also important aroma contributors, and organic acids, such as caproic acid, acetic acid, butyric acid and so on, are the main acids of Chinese strong-flavor liquor, giving it soft mouthfeel and long aftertaste.
We have added references ‘13, 14’. See lines64-69.
- Abstract and/or introduction should mention that high-throughput sequencing analysis is a DNA technique.
We have revised this content in Abstract and Introduction.
See lines 15, 53-62.
‘High throughput sequencing technology has been widely used in the study of microorganisms in pit mud and Zaopei of CSFL [7, 8, 9]. It is often used to analyze the influence of microbial diversity and community structure on liquor quality in brewing process. Shannon index and Chao1 index are usually used to reflect the richness and diversity of microorganisms. The higher Shannon index and Chao1 index are, the higher microbial richness and diversity are [10, 11]. The community structure can reflect the abundance of beneficial microorganisms in the pit mud and Zaopei. Through some statistical analysis methods, such as redundancy analysis (RDA) and Pearson correlation heatmap, correlation between environmental factors and microorganisms can be analyzed [9]. Therefore, we can use high-throughput sequencing to study the effect of pit mud on microorganisms and their changes in yellow water.’
We added the references ‘7, 8, 10, 11’.
- Introduction is not justifying some measured parameters as pH and acid analysis.
We have added this content in Introduction.
See line69-71: ‘Because of these organic acids, yellow water usually has higher acidity and lower pH value. This is a common index to judge the quality of yellow water [18].’
- Abstract:
Lines 15 Include DNA analysis.
This has been revised. See line 15.
- Lines 18 Although RDA is a known statistical abbreviation, the authors were using other statistical tools. For example, pearson correlation map is not mentioned in the experimental section.
We have added some content about statistical analysis, in Introduction, Materials and Methods, Result and Discussion.
See lines 58-60, 163, 272-274, 283-284.
- Keywords: I would substitute the word “Chinese strong-flavor liquor” by “white liquor” or “spirit drink”; similarly, other synonymous for “yellow water” should be instead included.
Chinese Baijiu can be classified into four basic aroma types, namely, light, soy sauce, rice and strong flavor-types. The strong flavor-type baijiu is different from other flavor types because it uses mud pit as the equipment for fermentation. The pit mud and yellow water are very important for its aroma characteristics. And this is not the case with other flavor liquors. So ‘Chinese strong-flavor liquor’ is more accurate than ‘white liqour’.
‘Yellow water’ is a special term in white liquor industry. So it's hard to replace it with other synonyms.
- Introduction:
Lines 35 Large amounts of aromas is not associated with a better wine quality. This idea is also repeated in line 284
In addition to the main flavor ingredients mentioned in the article, there are a lot of other ingredients in yellow water really. In this study, the main flavor components were studied, and we will continue to study the role of other flavor components in the follow-up study.
- 87-89 General objective should be rewritten.
We have rewritten the general objective. See lines 78-86.
‘The role of mud and its microorganisms in CSFL has been reported in a great deal of literature [9, 18], but the function of yellow water and its microbes is not clear. In order to know how the microbes change in this process, how the microbes in the yellow water interact with microbes in the mud, and how they contribute to the formation of aromas, the changes of the microbial community and main flavor components in the yellow water at different stages of late fermentation of CSFL and their relationship are studied in this paper. By this study, the theoretical basis will be laid for improving the quality of liquor by using yellow water, and new opportunities may be found for the comprehensive utilization of yellow water, which can improve the rate of high quality liquor and save cost for liquor factory.’
- Materials and methods:
The number of analyses and repetitions are not mentioned along this section in any parameter. Flavor components and their standards are not mentioned in this section.
The number of analyses and repetitions are three times for each sample. This has revised in the article. The internal standards: Butyl acetate and 2-ethylbutyric acid.
See lines: 105, 138-139.
- Lines112 Detection conditions are described from line 118
113 Reference 16 is only using the same chromatograph or is also applying GC conditions and/or MS conditions?
Because of the addition of references, the original Reference 16 now becomes Reference 22 which is applying GC conditions and MS conditions.
See lines 129,133.
- Results and discussion:
Lines156 In my opinion, absolute contents of aromas are not relevant in this study.
Odor activity value should be a better approach to have an idea of the aromatic influence.
Thank you very much for this very good advice. According to our detection conditions, the relative content of aromas is relatively accurate in this study. The absolute content of aromas is used to prove the change of relative content and it is not the key data.
Odor activity value is a very good approach to estimate the aromatic influence, and the absolute content is needed to calculate its value. We will try to use odor activity value estimating the aromatic influence in follow-up study.
- Lines 180 What do the authors mean with OTUs?
The mean is that we got 501different OTUs from all 12 samples, according to 97% similarity.
- Lines 182 What means Shanon and Chao 1 indexes? They are briefly described.
We have added this content in Introduction and the references ‘10, 11’.
See lines 55-57.
- Figures and tables:
Is possible to increase the letter size in Figure 5?
The letter size in Figure 5 has been increased. See Fig.5

Round 2
Reviewer 1 Report
- L66 – “In addition to these microorganisms…” – since the whole paragraph (L56-65) was added, it is no longer clear what microorganisms are concerned here.
- L147 – “Internal standard: Butyl acetate and 2-ethylbutyric acid” – in L119 authors stated that butyl acetate was used as an internal standard, it is confusing. If both compounds were used as internal standards, please add also the concentration of 2-ethylbutyric acid
- L332 – please remove the units from the pH values (mL Standard NaOH solution)
Also, some formatting errors occurred:
- L214 – chao -> Chao
- L240-242 – names of microorganisms in italic
- L282 – Class level -> class level
Author Response
We thank Reviewer 1 very much for your valuable suggestions again. We are very sorry for the fault of internal standard. We have reconfirmed the internal standard with the instrument operator in our lab again and revised the fault. Below is our point-by-point response to the comments.
- L66 – “In addition to these microorganisms…” – since the whole paragraph (L56-65) was added, it is no longer clear what microorganisms are concerned here.
We've deleted this sentence: ’In addition to these microorganisms, many kinds of flavor components of liquor are also in yellow water’ in L63-64 of the original manuscript. Then we’ve added the sentence:’ There are a lot of these acids and esters in yellow water [6]’ in L68-69. This makes the context seem more coherent.
See L63-64, 68-69.
- L147 – “Internal standard: Butyl acetate and 2-ethylbutyric acid” – in L119 authors stated that butyl acetate was used as an internal standard, it is confusing. If both compounds were used as internal standards, please add also the concentration of 2-ethylbutyric acid.
We have reconfirmed that the internal standard we used is Butyl acetate. According to our experience, the concentration of internal standard we used is 6.8mg/mL (chromatographic ethanol solution). On the basis of our detection conditions, the relative content of aromas is relatively accurate in this study. The absolute content of aromas is used to prove the change of relative content.
Actually we are trying to use 2-ethylbutyric acid as internal standard to test the accurate content of organic acids in fermented grains in follow-up study. Now we are testing the appropriate 2-ethylbutyric acid concentration as the internal standard.
See L138.
- L332 – please remove the units from the pH values (mL Standard NaOH solution)
The units from the pH values (mL Standard NaOH solution) has been removed.
See L321.
- L214 – chao -> Chao
Revised as request. See L205
- L240-242 – names of microorganisms in italic
This has been revised. See L231-233.
- L282 – Class level -> class level
This has been revised. See L273

This manuscript is a resubmission of an earlier submission. The following is a list of the peer review reports and author responses from that submission.
Round 1
Reviewer 1 Report
The manuscript is very interesting. After minor revision (the suggestions are contained in the attached file) the manuscript may be accepted for publication in the Fermentation journal.

Author Response
We want to begin by thanking Reviewer 1 for writing that “The manuscript is very interesting.” We also appreciated the constructive criticism and suggestion. We addressed all the points raised by the reviewer, as summarized below.
How was the acidity calculated?Take 1ml of yellow water mixed evenly and add it into a 250ml conical flask, add 50ml of water, titrate with 0.1mol/L NaOH (add 2 drops of phenolphthalein indicator first) to a reddish color, and take half a minute as the end point.
Acidity: for every 1ml of yellow water, 1ml of 0.1mol/L NaOH solution for titration is 1 degree.
Acidity=CV/0.1, C, NaOH solution concentration, mol/L; V, NaOH solution consumed in titration, mL; 0.1, Standard NaOH solution, mol/L.
See L96-L102 in manuscript.
Headspace adsorption conditions should be given.We have rewritten this part and added some adsorption conditions.
See L106-L123 in manuscript.
“We took 5 mL of yellow water in a headspace bottle, added 2 g of NaCl and 100 μL of the internal standard butyl acetate (8.4 mg/mL), equilibrated the mixture at 70 °C for 10 min, then extracted it by headspace adsorption for 35 min. Then the headspace needle was inserted into sample inlet of the gas chromatography mass spectrometer (GS-MS), and the sample was injected at 230 ℃, and the fragrance components were analyzed.”
“We took 2 g of mud in a headspace bottle, added 5 mL of deionized water and 2 g of NaCl, added 100 uL of the internal standard substance butyl acetate (8.4 mg/mL), and extracted the mixture for 30 min with the ultrasonic wave (Scientz IID ultrasonic cell grinder, power,250W, temperature, 25℃). The mixture was equilibrated at 70 °C for 10 min, then extracted it by headspace adsorption for 35 min. Then the headspace needle was inserted into sample inlet of the GS-MS, and the sample was injected at 230 ℃, and the fragrance components were analyzed.”
“The volatile component was extracted with a 50/30 μm Divinylbenzene/Carboxen/Polydimethylsiloxane (DVB/CAR/PDMS) extraction head (Supelco, Inc., Bellefonte, PA), which was aged at 270 ℃ for 1h to remove impurities. The instrument was MPS2 (Multipurpose sampler, automatic SPME) from Gerstel. The concentration was calculated by the percentage of the peak area.”
Ultrasonic wave technique should be described in more details?
We have added some ultrasonic wave technique conditions
“Scientz IID ultrasonic cell grinder, power, 250W, temperature, 25℃”.
See L114-L115 in manuscript.

Reviewer 2 Report
The manuscript „Changes of microbial community and aroma components in yellow water during the fermentation of Chinese strong-flavor liquor and the relationship with pit mud” describes the changes of the microbial community in yellow water and pit mud during a pit fermentation and their influence on the main aroma descriptors of Chinese liquor. This paper is quite interesting and it fits within the scope of the Fermentation journal but, in my opinion, needs to be improved. I have divided my comments into general and specific comments:
General comments:
In the introduction section, an explanation should be given why the authors focused only on short-chain fatty acids and their ethyl esters among the many aroma components Results for “day 0” should also be included Some minor English corrections should be doneSpecific comments:
L18 “Lcactobacillus”- should be Lactobacillus (same – L187) In the Materials and method section, more detailed description of pits used in experiments should be included (dimensions, covering). L63 – “before putting fermented grains into the pits” – what kind of grains were used? L67 – “pit mud was taken from two different sites (one site was on the pit wall and the other site on the pit bottom, near the yellow water sampling point in each pit)” – according to Fig S1, there are three sampling points. Could you please specify? L74 – what was the unit of acidity? Subsection 2.3. Flavor Components of Yellow Water and Pit Mud, should be rewritten as it is hard to follow the conditions used (e.g. the conditions of GC analysis are given twice (L81-81 and 95-97) and are different from each other; “Gas Chromatography-Mass Spectrometer (GC-MS) Requirement [11]” and “Agilent 6890N Gas Chromatograph - 5975 Mass Spectrometer” – what is this? Some headings?) L119 “vsesion7.0” – version 7.0 L126-128 – reference to supplementary materials is needed L129-153 – the title of the subsection suggests that complex analysis of the aroma compounds was performed, but authors focused only on 7 compounds (acetic acid, butyric acid, caproic acid, ethyl acetate, ethyl lactate, ethyl butyrate and ethyl caproate). Could you give an explanation of why did you select those compounds?Also, the reference to supplementary materials is needed when the concentration of individual compounds is described in the text.
Also, in the figure caption (Fig 1), why is abbreviation HS30, HS45 and HS60 included if it is not used in the figure?
L157 - reference to supplementary materials is needed Table 1 – the table footer should be included to explain the abbreviations used L166, 2x170 – “OUT” should be OUT L177 “…the microorganisms were definitely affected by the microbes” – please rewrite this part. Microorganisms and microbes is the same Figure 2A is very difficult to read, a sharper image is necessary L187-190 – the data presented in the text don’t match the ones presented in Figure 2B L193 – “table S2” should be S3 L197-201 – the same data are presented in Fig. 3A and in Table S3 Figure 3B is unreadable, a sharper image is necessary L229-230 – “The pH was positively correlated with Bacilli and negatively correlated with Clostridia (Fig. 4)” – the pH is not indicated in Fig 4 L242 “Lactobacillus was negatively correlated with the formation of these substances, and the most negative correlation was AA, followed by EA, BA, EL, CA, EC.” – According to Fig 5, the negative correlation of ethyl acetate was not so high Figure 5 needs to be sharper L252-259 – this paragraph should be in the Introduction section, as it clarifies the aim of the work L273-274 “At 30 days, alcoholic fermentation had been basically completed. The contents of ethanol and lactic acid in fermented grains and yellow water were relatively high [19].” – is this referred to own results of cited paper? L288 “This was consistent with the results in the Van diagram.” – Did you mean “Venn diagram”? Also, the reference to the appropriate figure is recommended.Author Response
Response to Reviewer 2 Comments
We would like to thank the Review 2 for their thoughtful review of our manuscript. We believe that the additional changes we have made in response to the reviewers comments have made this a significantly stronger manuscript. Below is our point-by-point response to the comments.
In the introduction section, an explanation should be given why the authors focused only on short-chain fatty acids and their ethyl esters among the many aroma components.
The main difference between Chinese liquor and other distilled liquors lies in the aroma of lower fatty acid ethyl ester and the high total acid content. CSFL mainly reflects the compound aroma dominated by ethyl caproate (EC). The complex aroma and flavor also includes ethyl acetate (EA), ethyl lactate(EL), ethyl butyrate(EB), caproic acid(CA), acetic acid(AA), butyric acid(BA), etc. In the lower part of pit, the fermented grains were immersed in yellow water for a long time. The content of these aroma components in yellow water could reflect their content in fermented grains to some extent. The flavor components are distilled into liquor by distillation of fermented grains, which affect liquor quality. The content and proportion of these aroma components determine the flavor type and quality of the wine. Manufacturers pay much attention to these, and guide producing according to these.
Results for “day 0” should also be includedAccording to production process and experience of Chinese strong-flavour liquor, there is no yellow water in pit at day 0. Yellow water is mainly formed in the alcohol fermentation period. According to the current production experience, the alcoholic fermentation stage is mainly carried out in the first 30 days. At the beginning of fermentation, little yellow water is produced, and there is little yellow water at the bottom of the pit. At about day 30, most of yellow water has been produced, and followed into the bottom of the pit. So we chose the yellow water on the 30th day as the starting sample.
Some minor English corrections should be doneWe have submitted this manuscript to MDPI's English editing service. They have helped us finish the English revision of the article.
L18 “Lcactobacillus”- should be Lactobacillus (same – L187)
See L18, L240 in manuscript.
In the Materials and method section, more detailed description of pits used in experiments should be included (dimensions, covering).Dimensions of pits, Length × width × dept: 3m × 2m × 1.8m,
Thickness of pit mud: about 10cm
See L83 in manuscript.
L63 – “before putting fermented grains into the pits” – what kind of grains were used?Fermented grains is generally composed of sorghum, rice, glutinous rice, corn and wheat in proportion.
See L34-L35 in manuscript.
L67 – “pit mud was taken from two different sites (one site was on the pit wall and the other site on the pit bottom, near the yellow water sampling point in each pit)” – according to Fig S1, there are three sampling points. Could you please specify?We have revised Fig S1 in which two sampling points have been re-annotated.
See Fig S1 in Supplementary Files.
L74 – what was the unit of acidity?Take 1ml of yellow water mixed evenly and add it into a 250ml conical flask, add 50ml of water, titrate with 0.1mol/L NaOH (add 2 drops of phenolphthalein indicator first) to a reddish color, and take half a minute as the end point.
Acidity: for every 1ml of yellow water, 1ml of 0.1mol/L NaOH solution for titration is 1 degree.
Acidity=CV/0.1, C, NaOH solution concentration, mol/L; V, NaOH solution consumed in titration, mL; 0.1, Standard NaOH solution, mol/L.
See L96-L102 in manuscript.
Subsection 2.3. Flavor Components of Yellow Water and Pit Mud, should be rewritten as it is hard to follow the conditions used (e.g. the conditions of GC analysis are given twice (L81-81 and 95-97) and are different from each other; “Gas Chromatography-Mass Spectrometer (GC-MS) Requirement [11]” and “Agilent 6890N Gas Chromatograph - 5975 Mass Spectrometer” – what is this? Some headings?)We have rewritten this part.
See L106-L136 in manuscript.
L119 “vsesion7.0” – version 7.0See L60 in manuscript.
L126-128 – reference to supplementary materials is neededWe have added supporting data. See Table S1 in Supplementary Files.
L129-153 – the title of the subsection suggests that complex analysis of the aroma compounds was performed, but authors focused only on 7 compounds (acetic acid, butyric acid, caproic acid, ethyl acetate, ethyl lactate, ethyl butyrate and ethyl caproate). Could you give an explanation of why did you select those compounds?We have explained the reasons for choosing these seven compounds. See L57-L68 in manuscript.
Also, the reference to supplementary materials is needed when the concentration of individual compounds is described in the text.We have added supporting data. See Table S2 in Supplementary Files.
Also, in the figure caption (Fig 1), why is abbreviation HS30, HS45 and HS60 included if it is not used in the figure?We have revised Fig 1 in which HS30, HS45 and HS60 have been used.
L157 - reference to supplementary materials is needed Table 1 – the table footer should be included to explain the abbreviations used.We have added supporting data. See Table S3 in Supplementary Files.
The table footer has been add to explain the abbreviations used in table 1. See L207-L208 in manuscript.
L166, 2x170 – “OUT” should be OUTSee L217-L221 in manuscript.
17 .L177 “…the microorganisms were definitely affected by the microbes” – please rewrite this part. Microorganisms and microbes is the same
This part has been rewritten. See L228-L230 in manuscript.
Figure 2A is very difficult to read, a sharper image is necessaryFigure 2A has been replaced by a sharper image.
L187-190 – the data presented in the text don’t match the ones presented in Figure 2BFigure 2B has been replaced by a right image.
L193 – “table S2” should be S3See L246 in manuscript.
L197-201 – the same data are presented in Fig. 3A and in Table S3Duplicate data in the original Table S3 has been deleted. See Table S4 in Supplementary Files.
Figure 3B is unreadable, a sharper image is necessaryFigure 3B has been replaced by a sharper image.
L229-230 – “The pH was positively correlated with Bacilli and negatively correlated with Clostridia (Fig. 4)” – the pH is not indicated in Fig 4Figure 4 has been replaced by a right image in which the pH is indicated.
L242 “Lactobacillus was negatively correlated with the formation of these substances, and the most negative correlation was AA, followed by EA, BA, EL, CA, EC.” – According to Fig 5, the negative correlation of ethyl acetate was not so highLactobacillus was negatively correlated with the formation of these substances, and the most negative correlation was AA, followed by BA, EL, CA, EC, then EB, EA.
See L308-L310 in manuscripts.
Figure 5 needs to be sharperFigure 5 has been replaced by a sharper image.
L252-259 – this paragraph should be in the Introduction section, as it clarifies the aim of the workThis paragraph has been revised and put in the Introduction section.
See L57-L68 in manuscript.
L273-274 “At 30 days, alcoholic fermentation had been basically completed. The contents of ethanol and lactic acid in fermented grains and yellow water were relatively high [19].” – is this referred to own results of cited paper?According to production experience, alcoholic fermentation had been basically completed and the content of ethanol should be relatively high after the grains fermented about 30 days. At this time lactic acid in fermented grains and yellow water was also higher (according to the reference [21]). See the L335-L339 in manuscript.
L288 “This was consistent with the results in the Van diagram.” – Did you mean “Venn diagram”? Also, the reference to the appropriate figure is recommended.Yes it’s “Venn diagram”. The Venn diagram is Figure 2A. See L353-L356 in manuscript.
